# Prognostic Clinical and Biologic Features for Overall Survival after Relapse in Childhood Medulloblastoma

**DOI:** 10.3390/cancers13010053

**Published:** 2020-12-27

**Authors:** Sophie Huybrechts, Gwénaël Le Teuff, Arnault Tauziède-Espariat, Caroline Rossoni, Anaïs Chivet, Émilie Indersie, Pascale Varlet, Stéphanie Puget, Rachid Abbas, Olivier Ayrault, Léa Guerrini-Rousseau, Jacques Grill, Dominique Valteau-Couanet, Christelle Dufour

**Affiliations:** 1Service National d’Oncologie et Hématologie Pédiatrique, Centre Hospitalier de Luxembourg, L-1210 Luxembourg City, Luxembourg; huybrechts.sophie@chl.lu; 2Department of Biostatistics, Gustave Roussy Cancer Center, Paris-Saclay University, 94800 Villejuif, France; gwenael.leteuff@gustaveroussy.fr (G.L.T.); caroline.rossoni@outlook.fr (C.R.); rachid.abbas@gustaveroussy.fr (R.A.); 3Department of Neuropathology, Sainte Anne Hospital, Rene Descartes University, 75014 Paris, France; a.tauziede-espariat@ghu-paris.fr (A.T.-E.); p.varlet@ghu-paris.fr (P.V.); 4Department of Pediatric Neurosurgery, Necker Hospital, Paris Descartes University, 75015 Paris, France; anais.chivet@aphp.fr (A.C.); stephanie.puget@aphp.fr (S.P.); 5Institut Curie, PSL Research University, CNRS UMR, INSERM, 91400 Orsay, France; emilie.indersie@curie.fr (É.I.); olivier.ayrault@curie.fr (O.A.); 6Paris Sud University, Paris-Saclay University, CNRS UMR 3347, INSERM U1021, 91400 Orsay, France; 7Department of Pediatric and Adolescent Oncology, Gustave Roussy, 94800 Villejuif, France; lea.guerrini-rousseau@gustaveroussy.fr (L.G.-R.); jacques.grill@gustaveroussy.fr (J.G.); dominique.valteau@gustaveroussy.fr (D.V.-C.); 8INSERM, Molecular Predictors and New Targets in Oncology, Paris-Saclay University, 94800 Villejuif, France

**Keywords:** recurrent medulloblastoma, molecular subgrouping, pediatric, outcome after relapse, time to relapse, salvage radiotherapy

## Abstract

**Simple Summary:**

Despite progress in the biology and upfront treatment of childhood medulloblastoma, relapse is almost universally fatal. No standardized treatment has so far been established for these patients. By determining which characteristics are prognostic after relapse, treatment strategies may be optimized for each of these children. We demonstrated that molecular subgroup at diagnosis is a relevant prognostic factor of outcome after relapse. Moreover, we showed that time to relapse and the use of salvage radiotherapy at relapse might have a potential impact on post-relapse survival. Our data suggest that ongoing efforts toward a better understanding of the biology, timing and type of relapse would be important to understand the determinants of tumor behavior at relapse. This could help us address more specific questions on the best surveillance strategies after completion of the treatment and the introduction of risk-stratified second-line treatment strategies.

**Abstract:**

Given the very poor prognosis for children with recurrent medulloblastoma, we aimed to identify prognostic factors for survival post-relapse in children with childhood medulloblastoma. We retrospectively collected clinico-biological data at diagnosis and main clinical characteristics at relapse of children newly diagnosed with a medulloblastoma between 2007 and 2017 at Gustave Roussy and Necker Hospital. At a median follow-up of 6.6 years (range, 0.4–12.3 years), relapse occurred in 48 out 155 patients (31%). The median time from diagnosis to relapse was 14.3 months (range, 1.2–87.2 months). Relapse was local in 9, metastatic in 22 and combined (local and metastatic) in 17 patients. Second-line treatment consisted of chemotherapy in 31 cases, radiotherapy in 9, SHH-inhibitor in four and no treatment in the remaining four. The 1-year overall survival rate post-relapse was 44.8% (CI 95%, 31.5% to 59.0%). While molecular subgrouping at diagnosis was significantly associated with survival post-relapse, the use of radiotherapy at relapse and time to first relapse (>12 months) might also have a potential impact on post-relapse survival.

## 1. Introduction

The management of medulloblastoma (MB) has evolved over the last three decades as a result of prospective multicentric clinical trials. Multimodal treatment including surgical resection, radiotherapy (RT) and chemotherapy (CT) has led to an improvement of outcomes with around two thirds of the patients being long-term survivors [1]. Treatment outcome is strongly associated with patient age and a series of established and evolving clinicopathological characteristics including metastatic disease, histology, postoperative residual disease (< or ≥1.5 cm^2^), *MYC* amplification status and more recent molecular features [2,3,4,5]. Five-year overall survival (OS) for standard-risk patients, typically defined as patients older than 3 years of age at diagnosis, who have a gross total resection of their tumor and are non-metastatic at diagnosis, ranges between 70–85% [6,7,8,9]. Patients who are younger than 3 years of age, have a subtotal resection (residual disease of >1.5 cm^2^) and/or are metastatic at diagnosis are considered to be “high risk” and often have a five-year OS <70% [10,11].

More recent insights into the biology of MB have shown that this group of tumors comprises at least four distinct molecular subgroups, Wingless (WNT), Sonic Hedgehog (SHH), Group 3 and Group 4, with transcriptionally and genetically distinct profiles and correlated clinical outcome. The WNT subgroup MB has an excellent prognosis wherein Group 3 MB carries the worst prognosis [3,4,5]. Currently, these molecular subgroups have been integrated into therapy stratification. For example, molecular features identified in international clinical trials (e.g., SJMB12 [NCT01878617] and PNET5 [NCT02066220]) have led to therapy reduction in groups with a good prognosis (e.g., MB WNT) and treatment intensification for groups at a high-risk of relapse.

Despite the advances in MB therapy, relapsed MB represents a major therapeutic challenge in pediatric neuro-oncology patients. Outcome after relapse remains variably poor with survival rates less than 10% at 5 years, except in patients who did not receive RT at diagnosis [12,13,14,15]. Interestingly, Ramaswamy et al. [16] showed that relapsed tumors retain their original molecular characteristics and that the pattern of relapse is subgroup dependent.

Although risk factors for OS in childhood MB have been well established at diagnosis, little is known about factors influencing survival at relapse. We aimed at a detailed analysis of clinico-biologically relevant factors at diagnosis and management at relapse to identify which factors are more prognostic of survival post-relapse. This might help guide therapeutic decisions and allow us to integrate these data into clinical trials at relapse.

## 2. Results

### 2.1. Patient Characteristics at Diagnosis

Table 1 shows the patients’ characteristics at diagnosis. A total of 155 patients (83 boys and 72 girls) were enrolled into this retrospective study with a median follow-up of 6.6 years (range, 0.4–12.3 years). The median age at diagnosis was 6.6 years (range, 0.1–18.4 years). Ninety-two children (59%) had localized disease (M0) while 63 (41%) had metastatic dissemination (M1–M3) at diagnosis.

According to the central histopathological review, the subtype of MB was classified as classic in 105 cases (68%), desmoplastic/nodular in 28 (18%), large cell/anaplastic (LCA) in 13 (8%) and not otherwise specified (NOS) in 9 cases (6%). No medulloblastoma with extensive nodularity was identified in this cohort.

DNA methylation-based subgrouping affiliation was available in 137 cases (88%). The most common subgroup was Group 4 (37%), followed by Group 3 (28%) and SHH tumors (24%). The WNT subgroup represented the smallest group (11%). Some histopathological features are clearly enriched in certain MB subgroups. Nearly all the WNT tumors had classical histopathology (13/15) with only one patient classified as LCA and one as NOS. All desmoplastic/nodular tumors (*n* = 28) were classified as belonging to the SHH-subgroup. Group 3 MB (*n* = 38) consisted of classic (26/38), LCA MB (8/38) and NOS in 4/38. Out of 51 Group 4 MB, 48 were classic MB, two were LCA and another one NOS. Metastatic disease was more commonly found at diagnosis in Group 3 (25/38; 66%) followed by Group 4 MB (21/51; 41%). For SHH tumors, metastasis was detected in 6 out of 33 cases (18%). All WNT MB had localized disease at diagnosis. Amplification of *MYC* occurred in 16/152 patients (11%): three *MYCN* amplification in SHH group five in Group 4 and data missing on molecular subgroup for one patient; five *MYC* amplification in Group 3 and two in Group 4.

Regarding first-line treatment, 122/155 (79%) patients received RT, either alone (*n* = 38; 25%) or in combination with CT (*n* = 84; 54%). Ten patients underwent focal RT and 103 craniospinal irradiation (CSI) with a median total CSI and total posterior fossa or posterior fossa tumor bed dose of 36 Gy (range, 18–36 Gy) and 54 Gy (range, 32–68 Gy), respectively. The remaining 9 patients had progressive disease prior to radiation therapy (Appendix A: patient # 21, 22, 29, 30, 32, 34, 37, 39 and 47).

The 5-year OS and EFS for the entire cohort were 77.2% (CI 95%, 69.6–83.4%) (*n* = 41 deaths) and 67.4% (CI 95%, 59.4–74.6%) (*n* = 54 events), respectively (Figure 1).

When comparing outcomes of patients according to molecular subgroups, WNT tumors had the best 5-year EFS and OS, followed by Group 4 and SHH MB, then Group 3. Of note, OS and EFS were not adjusted for differences in upfront treatment (RT-based regimen vs. CT only) (Figure 2).

### 2.2. Patients with Relapsed Medulloblastoma

At the last follow-up, relapse or progressive disease occurred in 48 patients (31%) at a median time of 14.3 months (range, 1.2–87.2 months) after diagnosis. A biopsy or surgical resection of the tumor was performed in case of focal relapse, in the context or molecular screening programs (precision medicine in the field of pediatric oncology) or in ambiguous cases. Table 1 describes the initial characteristics of the patients with relapsed MB. Compared to patients without relapse, patients with relapsed MB were generally younger than 5 years (*p* = 0.0115), with Group 3 MB (*p* = 0.0475) and had a *MYC/MYCN* amplification (*p* = 0.0248). The pattern of relapse was local in 9 cases, metastatic in 22 and combined local and metastatic in 17. Although all patients of Group 3 MB showed evident metastatic dissemination at relapse, no statistical association was found between molecular subgrouping at diagnosis and individual relapse patterns, i.e., local, metastatic only and combined, local and metastatic (*p* = 0.0769, Fisher’s exact test). However, when patients with metastatic relapse and those with both local and metastatic relapse were grouped together, a significant general association between molecular subgrouping at diagnosis and the pattern of relapse was observed (*p* = 0.0348) According to the first-line treatment (RT-based regimen vs. CT alone), there was a statistically significant difference in pattern of relapse (*p* = 0.007, Fisher’s exact test) with a higher proportion of local relapse in patients treated with chemotherapy alone, especially in SHH MB and Group 4 MB. Nearly all relapses were metastatic among the different molecular subgroups after receiving upfront RT (Figure 3).

Time to relapse was significantly dependent on molecular subgrouping (*p* < 0.0001, Kruskal-Wallis test). The median time to relapse from diagnosis for Group 3 MB was 0.66 years (range, 0.1–2.0 years), whereas the median time to relapse for SHH, WNT and Group 4 MB were 1.29 years (range, 0.51–7.27 years), 1.53 years (range, 1.33–1.73 years) and 2.08 years (range, 0.84–4.35 years), respectively. When looking at the time to relapse according to the treatment received at diagnosis, we observed that 20 out of 48 relapses occurred quickly within one year after diagnosis: 5/13 (38%) in the chemotherapy only group and 15/35 (43%) after treatment with upfront radiotherapy (Appendix A).

Salvage treatment modalities were selected according to initial treatment and on case-to-case basis. Fifteen out of 48 patients were able to undergo resection or biopsy of the relapsing tumor which involved local recurrence in nine cases and metastatic sites in six. Among the 35 patients who received upfront RT, re-irradiation was administered in three patients with (*n* = 1) or without chemotherapy (*n* = 2), two patients received an SHH-inhibitor (Appendix A: patient # 43 was initially considered as SHH MB according to IHC and subsequently categorized as Group 3 by DNA methylation), 26 were treated with chemotherapy and four patients did not receive further treatment. Thirteen patients progressed or relapsed after treatment based on chemotherapy alone: three received an SHH-inhibitor, four salvage chemotherapy and six patients RT with (*n* = 3) or without chemotherapy (*n* = 3). The detailed characteristics of the 48 relapsed patients are shown in the Appendix A. Figure 4 represents a flowchart of the entire study cohort.

Figure 5 and Figure 6 represent upfront treatment according to DNA-methylation subgroups in patients with relapsed medulloblastoma (Figure 5) and treatment at relapse according to DNA-methylation subgroups in patients with relapsed medulloblastoma after receiving upfront radiotherapy-containing regimen and chemotherapy only (Figure 6).

The median follow-up for patients with relapse/progression was 5.6 years (range, 0.5–10.0 years). The median OS from first relapse/progression was 0.91 years (CI 95%, 0.31–1.49 years) (37 deaths), with an estimated 1-, 2- and 5-year OS rate from first relapse of 44.8% (CI 95%, 31.5–59.0%), 32.5% (CI 95%, 20.5–47.3%) and 19.3% (CI 95%, 9.9–34.0%), respectively (Figure 7).

At the last follow-up, 11 out of 48 relapsed patients were alive with a median follow-up of 3.13 years (range; 0.5–10 years) (Appendix A). In these 11 children, relapse/progressive disease occurred at a median time after diagnosis of 2.02 years (range; 0.51–3.95 years). Recurrence occurred on first-line treatment only in one patient out of 11 (patient # 4). A relapse was confirmed by biopsy in 6 out of 11 cases. The pattern of relapse was isolated to the primary site in three cases (27%), metastatic in six (54%), and combined (local and metastatic) in two (18%). Among these 11 patients, five patients were initially treated with chemotherapy alone and subsequently treated at relapse with a multimodal treatment with RT (mode 36/54 Gy; range 25.4–36/50.4–54 Gy) in four cases and SHH-inhibitor therapy in one patient. For the remaining six patients who underwent radiotherapy at diagnosis, salvage treatment consisted of chemotherapy based on Temozolomide in five cases and SHH-inhibitor for the last one.

### 2.3. Second Malignant Neoplasms (SMN)

In our cohort, we observed six secondary malignancies: one acute leukemia, four fossa posterior high-grade gliomas and one hepatic tumor. These patients have been referred for genetic counseling and underwent germline testing. Of the six patients, only one was found to carry a germline TP53 mutation.

### 2.4. Prognostic Factors of Overall Survival after Relapse

Table 2 summarizes results of the penalized full multivariable Cox descriptive core model and its extensions. As further described below in Materials and Methods, we first included treatment, DNA methylation subgrouping, age, extend of the disease (M-stage) at diagnosis and treatment at relapse into the core descriptive model. Only DNA methylation subgrouping was statistically significantly associated to OS post-relapse (*p* = 0.0021) with the highest risk of death for Group 3 MB (adjusted HR 13.009; 95% CI, 1.437–117.757) (all patients of Group 3, *n* = 18, died). The risk of death was approximately the same for patients in Group 4 and SHH (approximately 1.5 times higher than in WNT patients). Although not statistically significant at the 5% level, there was a marginal effect (*p* = 0.0910) of RT at relapse; patients receiving RT at first relapse had a decreased risk of death (adjusted HR 0.350; 95% CI, 0.104–1.182). Similar results have been observed when adding *MYC* status to the core descriptive model. We then added the time to first recurrence into the descriptive core model, and noticed that DNA methylation subgrouping remained significantly associated to OS post-relapse (*p* = 0.0190) and RT at first relapse became significantly associated with a better post-relapse survival with adjusted HR = 0.203, 95% CI [0.055–0.752] (*p* = 0.0170).

To assess whether time interval between diagnosis and first relapse is associated with survival after relapse, we first evaluated its functional form. The assumption of log linearity was violated (*p* = 0.0370, univariate analysis). The cut-off was defined after performing an analysis by the quartiles showing a lower risk of death after a median value of 12 months (data not shown). The group of children who relapsed later than one year had a better prognosis (HR 0.289; 95% CI, 0.098–0.846, *p* = 0.0235). Similar results were obtained, adding *MYC* status again to the previous variables (last column of Table 2).

Due to the small number of patients in these analyses and possible instability of our results, we evaluated the robustness of these findings by a bootstrap analysis. The principle consists in repeating the different analyses on several samples (B = 5000 chosen arbitrarily) drawn by random resampling with replacement from the original database, and computing the percentage at which each variable within a model is statistically significant at a significance level of 5%. Based on an arbitrary cut-off of 60%, referring to the Sauerbrei’s work [17] on variables selection, this method confirmed that molecular subgrouping by DNA methylation is a prognostic factor of OS post-relapse with a percentage of selection higher than 67%. The percentage of selection of salvage RT at first relapse varied between 39% to 59%, and that of the time between diagnosis and first relapse was less than 56%. As such, no definitive conclusion could be drawn about the prognostic value of both explanatory variables on OS post-relapse (Table 3).

## 3. Discussion

Limited studies have been published on the outcome post-relapse and the impact of clinico-biological data at diagnosis and relapse on post-relapse survival in childhood MB. This large retrospective study, focusing on putative clinico-molecular factors associated with post-relapse survival in childhood, identified molecular subgrouping at diagnosis to be significantly related to survival post-relapse. Compared to the WNT subgroup, patients in Group 3 had a significant greater risk of death at relapse, followed by Group 4 and SHH tumors. The poor outcome of Group 3 MBs results from a coherence of different unfavorable prognostic factors at diagnosis such as young age, LCA histologic subtype, metastatic dissemination and the presence of MYC amplification [3,4,5,18]. Similar to previous studies [16], poor survival has been observed in Group 3 MBs even after relapse.

While time to relapse has been established to be correlated with survival post-relapse for pediatric patients with relapsed malignancies, it has not been thoroughly studied in children with relapsed MB [19,20,21,22]. The present study showed a trend toward a better outcome after relapse when it occurred more than 12 months after diagnosis. A relevant question today is whether certain clinico-biological factors at diagnosis may predict an early relapse, less than 12 months from diagnosis. From what we observed in our study population at relapse, was that Group 3 MB patients had a shortened median time to relapse followed by SHH, WNT and Group 4 MBs, with the majority of Group 3 patients relapsing within a time interval of one year from diagnosis. Nearly all of them received upfront radiotherapy. Similar results have been published by Ramaswamy et al. [16], who found a significant difference in time to relapse between the different molecular subgroups. They demonstrated that Group 4 MBs relapsed significantly later than the other subgroups and are shown to survive longer after relapse, irrespective of treatment regimens received at diagnosis. These findings strengthen the need of a better understanding of tumor biology at diagnosis and relapse. In a recent study, a similar significant reduction in time to relapse was reported in previously irradiated Group 3 MB, more particularly in subtype III after assignment into novel second-generation molecular subtypes within Group 3 MB [18].

The treatment at relapse remains challenging and different approaches exist including re-resection, re-irradiation, a variety of chemotherapeutics, high-dose chemotherapy (HDCT) with autologous stem cell transplantation (ASCT) and novel targeted therapies [23,24,25,26,27,28,29]. While radiation therapy and the addition of chemotherapy has significantly improved the rate of survival in children older than 3–5 years with both average- and high-risk MB, there remains considerable debate as to whether the use of irradiation at the time of relapse may improve OS. Previous studies provided evidence on the potential effectiveness of radiation therapy as a salvage treatment in a subset of patients, for example in young children previously treated with radiation sparing regimen, albeit at the cost of significant neuropsychological deficits and other severe sequelae [18,25,30]. Although our analysis lacked statistical power, we found that salvage radiotherapy might have a potential impact on the outcome after relapse, regardless of upfront treatment. Of note, among the nine patients who benefit from RT at relapse, four patients, still alive at the last follow-up, have been treated at first- line with chemotherapy alone.

It is noteworthy that in our study, age, *MYC* status, treatment regimen and metastatic stage at diagnosis were not significantly associated with post-relapse survival. The small numbers of patients in our cohort could have biased the prognostic significance of *MYC* status. Earlier published studies on survival after relapse have only shown no effect of metastatic dissemination on outcome [14,15].

In the present study, the pattern of relapse was not related with the molecular subgroup. Ramaswamy et al. [16] identified differences in relapse patterns among the four molecular subgroups, independent of treatment at diagnosis. More local relapses in patients with SHH tumors compared to Group 3 and Group 4 tumors, which tended to recur with metastatic dissemination, suggesting that subgroup affiliation, rather than treatment effects seems to be the primary driver of location of relapse. In our study, among the nine patients with relapsed SHH MB, six had a metastatic or combined local and metastatic relapse. In a more recent study [18], distant relapses were also predominant in patients with SHH MB. Although recurrence rates are low in WNT subgroup, two of our WNT patients had recurrent metastatic disease and were treated at diagnosis with RT.

We should underline some limitations of the current study. First, this retrospective study cannot exclude any bias. Second, the small number of patients with a relapse limited our exploration of more explanatory variables possibly associated to OS post-relapse and more particularly limited our ability to draw any conclusion on the prognostic value of salvage radiation therapy at relapse and the time to relapse on OS post-relapse. Another limitation is the lack of information on TP53 mutation status at diagnosis in SHH MB. Finally, based on the variability in therapies used in this study, we cannot comment on the relative benefits of different chemotherapy salvage regimens.

## 4. Materials and Methods

### 4.1. Patients Population

The medical records of 155 consecutive MB, aged 18 years or below, and diagnosed at Gustave Roussy and Necker University Hospital between 2007 and 2017, were reviewed. Metastatic disease was assessed using the staging system described by Chang et al. [31].

Parents/guardians gave written informed consent for the retrospective analysis of clinical data according to the Institutional Review Board (IRB) and before inclusion into ongoing protocols.

### 4.2. Subgroup Determination

Two neuropathologists reviewed all available tumor samples according to the 2016 World Health Organization (WHO) guidelines [32]. Tumor DNA was extracted with the Qiagen DNeasy Blood & Tissue kit (Cat NO./ID 69504) according to the manufacturer’s instructions from freshly frozen tissue samples. For DNA profiling, 500 ng total DNA were sent to the Genotyping facility at the German Cancer Research Center (Heidelberg, Germany). All patient samples were analyzed using either Illumina Infinium Methylation EPIC or HumanMethylation450 BeadChip arrays in accordance with manufacturer’s instructions. MB subgroup affiliation predictions were obtained from a DNA methylation-based classification web-platform for central nervous system tumors (www.molecularneuropathology.org, version 11b4). Although TP53 mutation was associated with significant poorer outcomes in SHH MB, determination of TP53 mutation was not available in our study.

### 4.3. Treatment Regimens

After initial diagnosis, patients were treated according to different clinical trials. The treatment varied with time and was stratified according to risk factors such as histology, metastatic disease, postoperative residual disease and *MYC* status at diagnosis. Patients younger than 5 years of age commonly received radiation-sparing treatment [33,34,35,36,37,38] whereas children older than 5 years of age received conventional multi-modal therapy which included CSI [8,9,39]. For the statistical analyses, we classified patients into two groups of treatment regimens at diagnosis: patients treated with CT only and those treated with regimens containing RT.

### 4.4. Relapse

Relapse or progressive disease were confirmed by central review of Magnetic Resonance Imaging (MRI) scans. A biopsy was performed in ambiguous cases.

### 4.5. Statistical Analysis

The date of initial diagnosis was defined as the date of initial surgery. The median follow-up time from diagnosis was estimated using Schemper’s method [40]. OS and OS post-relapse were defined as the time from diagnosis and from first progression/relapse, respectively to death from any cause or to the date of the last contact for survivors. Event-free survival (EFS) was defined as the time from diagnosis to relapse, progression, second malignancy or death resulting from any cause. OS, OS post-relapse and EFS were estimated using the Kaplan–Meier method. Relapse was defined as patients developing local and/or metastatic disease after response to prior therapy, while an increase in the tumor’s size or the detection of new lesions was considered as tumor progression. Time to relapse was calculated as time from study enrollment to first occurrence of relapse or progressive disease. Comparisons of patients’ characteristics between the subgroup of patients who progressed or relapsed and those who did not, were performed using a Chi-Square or Fisher’s exact test.

To assess the prognostic effect of pre-specified variables on OS post-relapse, a full Cox proportional hazard regression model was used with no variable selection procedure since model selection strategies apply to small samples produce unreliable and unstable models [41,42]. A descriptive core model was constructed including the following factors: treatment at diagnosis (CT only vs. RT-based regimen), DNA methylation subgroup (WNT, SHH, Group 3, Group 4), age (< or ≥5 years), M-stage (M0, M+) and RT at first relapse (no, yes). Treatment at diagnosis is known to be correlated to OS, the three other variables are established prognostic factors of OS and second-line RT might be a salvage treatment for patients for whom irradiation was omitted at diagnosis. Due to a small number of patients with relapse, we then choose to add, separately and then together, the *MYC* amplification status and the time interval between diagnosis and first relapse into the core model, to assess their prognostic value on OS post-relapse. We did not choose to integrate “histology” as a variable to the core descriptive model due to its well-known and strong correlation with DNA methylation. To estimate the adjusted hazard ratios (HR) and its 95% confidence interval (CI), we used a multivariable Cox model with a Firth’s penalized maximum likelihood approach [43] as the standard maximum likelihood does not work well with estimation biased away from zero in small datasets [41,42].

We assessed the functional form of the time interval between diagnosis and first relapse by using a residual-based approach [44]. The hypothesis of proportionality hazard of the Cox model was evaluated by the Grambsch–Therneau test and no violation was observed (data not shown) [45]. A sensitivity analysis evaluating the stability of the associations between variables and OS post-relapse was performed using the bootstrap technique [17,46]. Two-sided *p*-values are reported and a *p*-value ≤ 0.05 was considered as statistically significant. The last follow-up data was updated on April, 3 2020. Statistical analyses were performed using SAS version 9.4 (SAS Institute Inc. Cary, NC, USA).

## 5. Conclusions

While survival of newly diagnosed childhood MB has been improved in the last decades, the outcome of patients with recurrent MB remains poor. Our study demonstrated the prognostic effect of molecular subgroup on survival post-relapse. Moreover, we showed that the use of salvage radiotherapy at relapse and the time between diagnosis and relapse of more than 12 months might have a potential impact on post-relapse survival. These findings could influence prognostication post-relapse and help design specific clinical trials for relapsed MB patients.

## Figures and Tables

**Figure 1 cancers-13-00053-f001:**
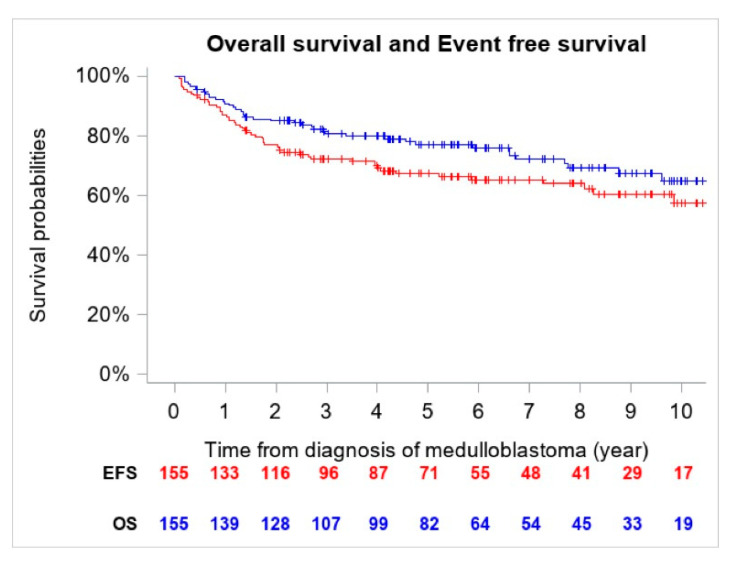
Overall Survival (OS) and Event Free Survival (EFS) of the entire cohort (*n* = 155).

**Figure 2 cancers-13-00053-f002:**
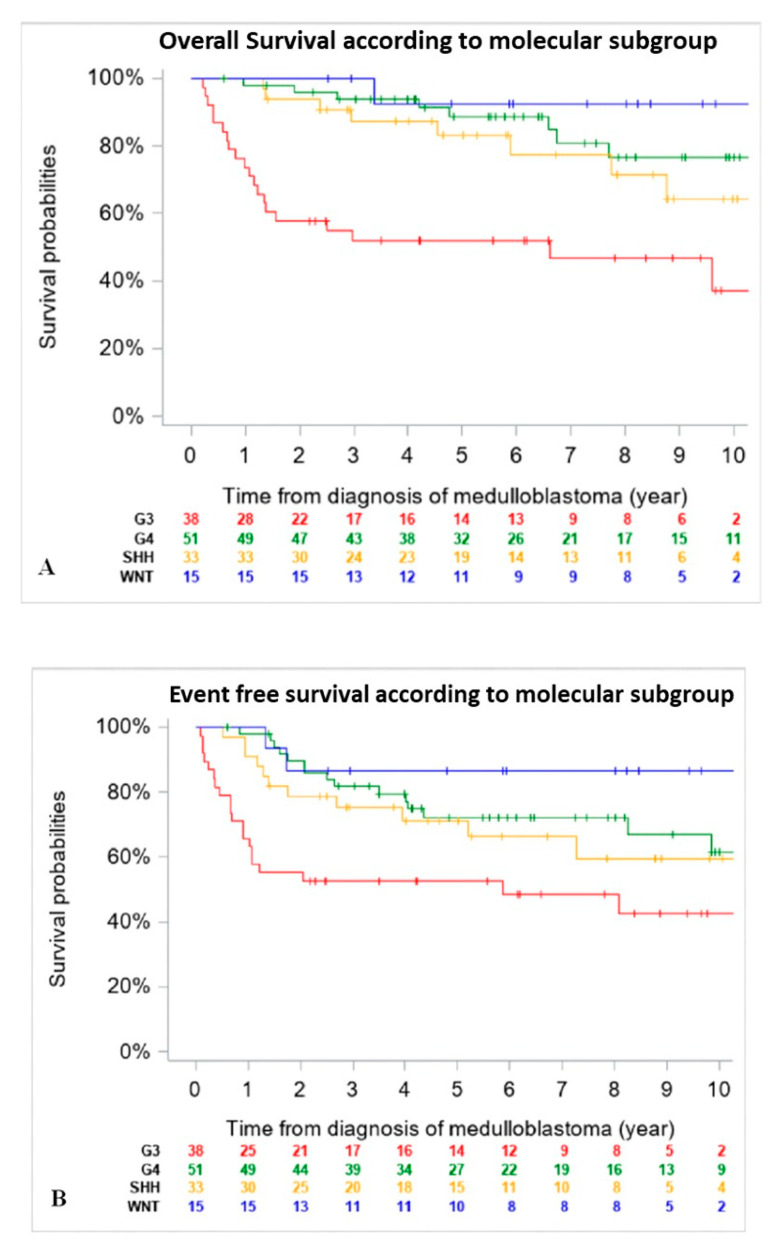
Overall Survival (OS) (**A**) and Event Free Survival (EFS) (**B**) of the entire cohort according to molecular subgroup (*n* = 137) (18 patients were excluded due to missing data on DNA methylation subgroups at diagnosis). (**A**): 5-year OS was 92.3% (CI 95%, 66.7–98.6%) for patients with WNT tumors, 83% (CI 95%, 65.6–92.6%) for SHH tumors, 88.6 (CI 95%, 75.7–95.0%) for Group 4 and 51.8% (CI 95%, 36.3–67.0%) for Group 3 MB. (**B**): 5-year EFS was 86.7% (CI 95%, 62.1–96.3%) for patients with WNT tumors, 71.3% (CI 95%, 53.6–84.2%) for SHH tumors, 72.2% (CI 95%, 57.9–83.1%) for Group 4 and 52.6% (CI 95%, 37.3–67.5%) for Group 3 MB.

**Figure 3 cancers-13-00053-f003:**
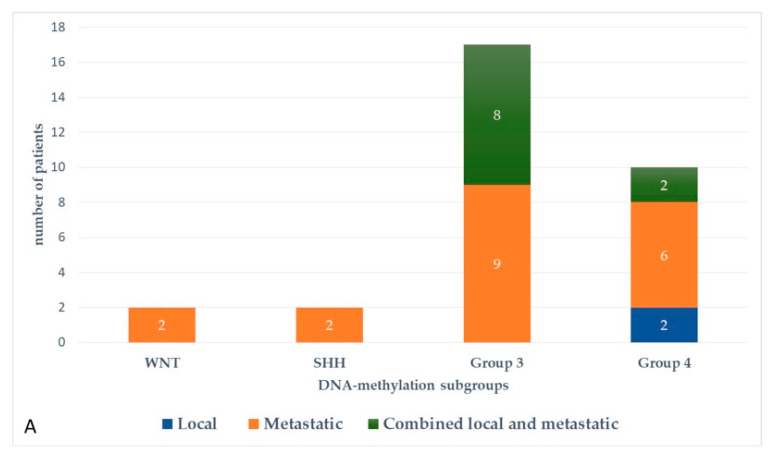
Pattern of relapse according to DNA-methylation subgroups in patients with relapsed medulloblastoma after receiving upfront radiotherapy-containing regimen (**A**) and chemotherapy only (**B**).

**Figure 4 cancers-13-00053-f004:**
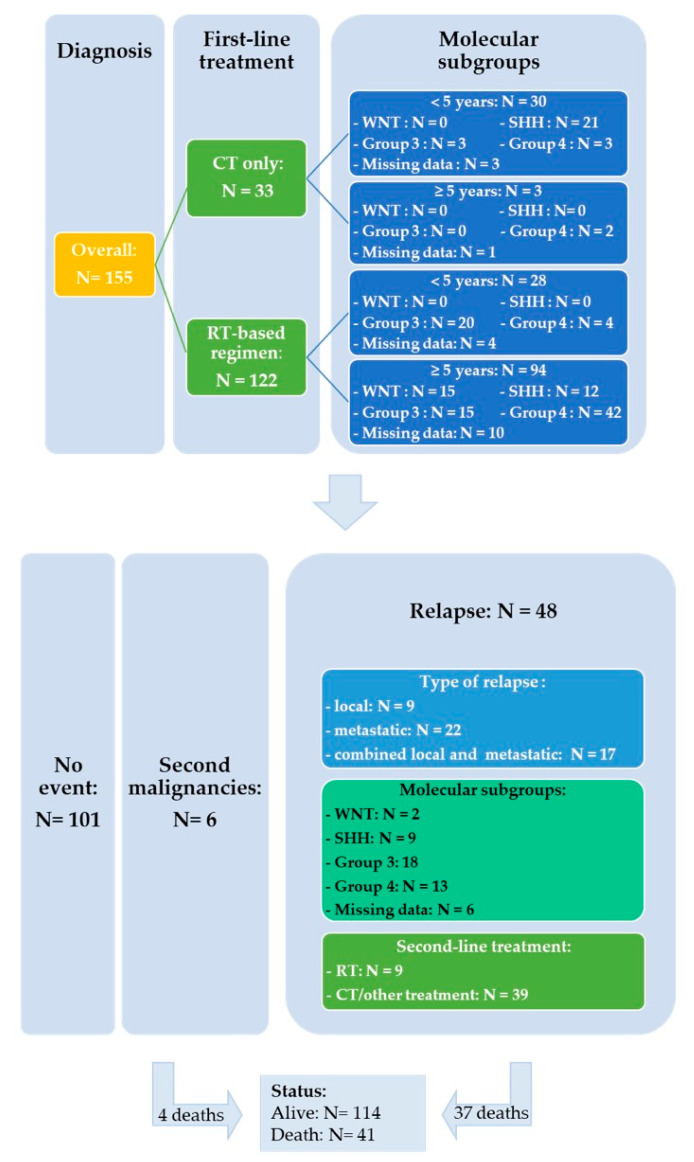
Flowchart of the entire study cohort.

**Figure 5 cancers-13-00053-f005:**
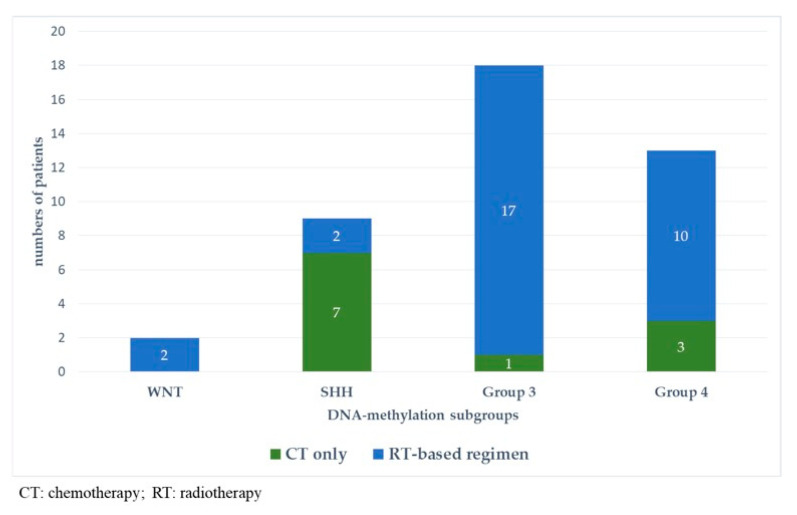
Upfront treatment according to DNA-methylation subgroups in patients with relapsed medulloblastoma.

**Figure 6 cancers-13-00053-f006:**
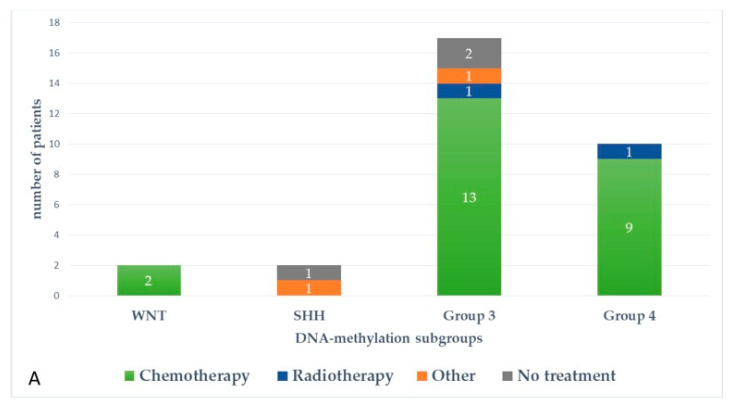
Treatment at relapse according to DNA-methylation subgroups in patients with relapsed medulloblastoma after receiving upfront radiotherapy-containing regimen (**A**) and chemotherapy only (**B**).

**Figure 7 cancers-13-00053-f007:**
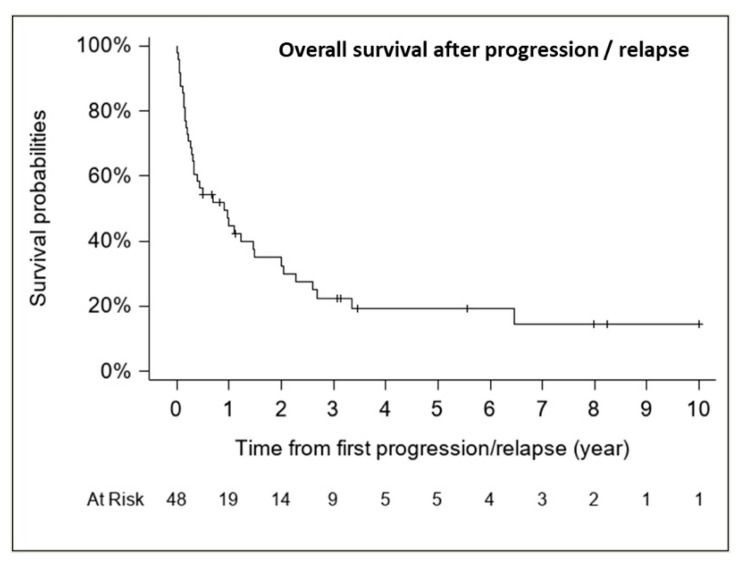
Overall survival after progression/relapse (*n* = 48, 37 deaths).

**Table 1 cancers-13-00053-t001:** Clinical characteristics at diagnosis and relapse.

	Patients at Diagnosis (*n* = 155)	Patients with a Relapse (*n* = 48)	Patients without Relapse (*n* = 107)	*p*-Value *
Age at diagnosis <5 years ≥5 years	58 (37%) 97 (63%)	25 (52%) 23 (48%)	33 (31%) 74 (69%)	0.0115
Sex Male Female	83 (54%) 72 (46%)	27 (56%) 21 (44%)	56 (52%) 51 (48%)	0.6515
Histology at diagnosis Desmoplastic/nodular Classic LCA NOS	28 (18%) 105 (68%) 13 (8%) 9 (6%)	9 (19%) 29 (60%) 7 (15%) 3 (6%)	19 (18%) 76 (71%) 6 (6%) 6 (6%)	0.2783
DNA methylation subgroups at diagnosis WNT SHH Group 3 Group 4 Missing data	15 (11%) 33 (24%) 38 (28%) 51 (37%) 18	2 (5%) 9 (21%) 18 (43%) 13 (31%) 6	13 (14%) 24 (25%) 20 (21%) 38 (40%) 12	0.0475
M-stage at diagnosis M0 M1 M2 M3	92 (59%) 4 (3%) 18 (12%) 41 (26%)	24 (50%) 1 (2%) 6 (13%) 17 (35%)	68 (64%) 3 (3%) 12 (11%) 24 (22%)	0.3296
*MYC/MYCN* amplification No Yes Missing	136 (89%) 16 (11%) 3	39 (81%) 9 (19%) 0	97 (93%) 7 (7%) 3	0.0248
Treatment at diagnosis CT-based only RT-based	33 (21%) 122 (79%)	13 (27%) 35 (73%)	20 (19%) 87 (81%)	0.2380
Treatment regimens at diagnosis CT alone CT-HDCT RT alone RT – CT CT-HDCT-RT	24 (15%) 9 (6%) 38 (25%) 13 (8%) 71 (46%)	10 (21%) 3 (6%) 4 (8%) 3 (6%) 28 (58%)	14 (13%) 6 (6%) 34 (32%) 10 (9%) 43 (40%)	0.0215

Abbreviations: LCA: Large cell/Anaplastic; NOS: Not Otherwise Specified; WNT: Wingless; SHH: Sonic Hedgehog; CT: chemotherapy; RT: radiotherapy; HDCT: High dose chemotherapy. *: Chi2 or exact Fisher’s test for comparison of patients’ characteristics between patients with and without relapse.

**Table 2 cancers-13-00053-t002:** Penalized (Firth’s approach) full Cox regression analysis for overall survival post recurrence (*n* = 48, 37 deaths) (origin time is the date of first progression/relapse) ^†^,*.

Characteristics	# Deaths/ # Patients	Descriptive Core Model		Descriptive Core Model + *MYC* Status		Descriptive Core Model + Time between Diagnosis and 1st Relapse		Descriptive Core Model + *MYC* Status + Time between Diagnosis and 1st Relapse	
		HR (95% CI)	*p*-Value	HR (95% CI)	*p*-Value	HR (95% CI)	*p*-Value	HR (95% CI)	*p*-Value
Treatment at diagnosis CT-based only RT-based	8/13 29/35	1 0.826 [0.233–2.932]	0.7676	1 0.752 [0.199–2.836]	0.6735	1 0.875 [0.242–3.164]	0.8387	1 0.841 [0.221–3.192]	0.7988
DNA methylation at diagnosis WNT SHH Group 3 Group 4	1/2 6/9 18/18 8/13	1 1.437 [0.154–13.391] 13.009 [1.437–117.757] 1.749 [0.246–12.446]	0.0021	1 1.528 [0.158–14.742] 14.682 [1.560–138.158] 1.655 [0.229–11.974]	0.0022	1 1.746 [0.178–17.180] 12.673 [1.349–119.017] 2.514 [0.348–18.147]	0.0190	1 1.782 [0.178–17.872] 13.293 [1.362–129.711] 2.462 [0.335–18.081]	0.0228
Age at diagnosis <5 year ≥5 year	21/25 16/23	1 1.009 [0.361–2.820]	0.9866	1 1.133 [0.381–3.375]	0.8221	1 1.410 [0.482–4.128]	0.5304	1 1.464 [0.481–4.460]	0.5023
M-stage at diagnosis M0 M+	17/24 20/24	1 1.074 [0.423–2.728]	0.8810	1 0.902 [0.319–2.553]	0.8464	1 0.731 [0.280–1.909]	0.5228	1 0.693 [0.246–1.947]	0.4863
Radiotherapy at 1st relapse No Yes	32/39 5/9	1 0.350 [0.104–1.182]	0.0910	1 0.327 [0.095–1.130]	0.0774	1 0.203 [0.055–0.752]	0.0170	1 0.198 [0.052–0.756]	0.0178
*MYC/MYCN* amplification at diagnosis No Yes	29/39 8/9			1 1.575 [0.574–4.325]	0.3778			1 1.208 [0.441–3.311]	0.7135
Time between diagnosis and 1st relapse * ≤1 year >1 year	19/20 18/28					1 0.289 [0.098–0.846]	0.0235	1 0.295 [0.099–0.883]	0.0291

Abbreviations: CT: chemotherapy; RT: radiotherapy; HR: Hazard Ratio, CI: confidence interval, *: the functional form of the time between diagnosis and 1st relapse was assessed by using a residual-based approach in a univariate analysis, ^†^: Data analysis was based on 42 patients as 6 patients were excluded because of missing data on DNA methylation.

**Table 3 cancers-13-00053-t003:** Robustness analyses based on bootstrap resampling.

Characteristics	Descriptive Core Model	Descriptive Core Model + *MYC* Status	Descriptive Core Model + Time between Diagnosis and 1st Relapse	Descriptive Core Model + *MYC* Status + Time between Diagnosis and 1st Relapse
Treatment at diagnosis	8.84	9.92	7.62	8.66
DNA methylation at diagnosis	90.24	90.84	67.7	67.32
Age at diagnosis	5.66	5.4	11.66	12.52
M-stage at diagnosis	4.96	9.2	10.88	15.16
Radiotherapy at 1st relapse	39.42	42.42	59.82	59.74
*MYC/MYCN* amplification at diagnosis		14.7		11.14
Time between diagnosis and 1st relapse			55.38	52.68

## Data Availability

The data on relapse presented in this study are available in Appendix A.

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
