# Peer review of "Prognostic Clinical and Biologic Features for Overall Survival after Relapse in Childhood Medulloblastoma"

_cancers, 2020, doi:10.3390/cancers13010053_

Round 1
Reviewer 1 Report
In this manuscript, Huybrechts et al., reported a retrospective study regarding children with medulloblastoma. In this work, the total of 155 patients were collected in the cohort that includes four molecular-subtypes. Besides describing the clinical features of this cohort, the authors also analyzed DNA-methylation, tumor metastases, and treatment response.
Although this study does not provide new information about medulloblastoma, the retrospective analysis supports the current knowledge of this brain tumor.
Here are a few comments below: 1) the authors described that all group 3 tumors have metastatic dissemination at relapse and only 31% patients had relapse, but no statistical association between subtypes and relapse. How to explain this point? what are the data to support it?
2) what are the differences of DNA-methylation profiles between local and metastatic tumors in each subtypes of medulloblastoma?
Author Response
Response to Reviewer 1 Comments
The authors would like to thank the reviewer for precise and thoughtful comments and constructive suggestions which has led to a better manuscript. Below we respond to each referee comments individually. Please note that lines and pages refer to the revised manuscript with track changes. Please see the attachment for a better overview of the tables.
Point 1: The authors described that all group 3 tumors have metastatic dissemination at relapse and only 31% patients had relapse, but no statistical association between subtypes and relapse. How to explain this point? what are the data to support it?
Response 1:
This point may be explained by the definition of “metastatic dissemination”, used in this sentence. We wrote in the paper that “Although all patients of Group 3 MB presented evident metastatic dissemination at relapse, we found no general statistical association between molecular subgrouping at diagnosis and the pattern of relapse (p = 0.0769, Fisher’s exact test)”. In this association, we distinguished patients with metastatic relapse only (n=20) from patients with combined, local and metastatic relapse (n=15). The table below describes in more detail this information.
|
Subtypes at diagnosis |
Type of relapse (n=48) |
|||
|
|
Tumor bed (local) |
Metastatic only |
Local and metastatic |
Total |
|
WNT |
0 (0.0%) |
2 (100.0%) |
0 (0.0%) |
2 |
|
SHH |
3 (33.3%) |
3 (33.3%) |
3 (33.3%) |
9 |
|
G3 |
0 (0.0%) |
9 (50.0%) |
9 (50.0%) |
18 |
|
G4 |
4 (30.8%) |
6 (46.2%) |
3 (23.1%) |
13 |
Fisher exact test p-value: 0.0769 since the Chi2 test may not be valid because of the expected counts less than 5. Six patients with missing data.
Under the hypothesis that the data from patients with metastatic relapse only and patients with combined, local and metastatic relapses, are grouped together, then the table will be as follows:
|
Subtypes at diagnosis |
Type of relapse (n=48) |
||
|
|
Tumor bed (local) |
Metastatic / Local and metastatic |
Total |
|
WNT |
0 (0.0%) |
2 (100.0%) |
2 |
|
SHH |
3 (33.3%) |
6 (66.7%) |
9 |
|
G3 |
0 (0.0%) |
18 (100.0%) |
18 |
|
G4 |
4 (30.8%) |
9 (69.2%) |
13 |
Fisher exact test p-value: 0.0348 since the Chi2 test may not be valid because of the expected counts less than 5. Six patients with missing data.
The general association between molecular subtypes at diagnosis and pattern of relapse in patients with relapsed medulloblastoma (n=48) becomes now statistically significant with p<0.05. In consequence, we modified the sentence as follows, page 6, line 144-150:
“Although all patients of Group 3 MB showed evident metastatic dissemination at relapse, no statistical association was found between molecular subgrouping at diagnosis and individual relapse patterns i.e. local, metastatic only and combined, local and metastatic (p = 0.0769, Fisher’s exact test). However, when patients with metastatic relapse and those with both local and metastatic relapse were grouped together, a significant general association between molecular subgrouping at diagnosis and the pattern of relapse was observed (p=0.0348).”
Point 2: What are the differences of DNA-methylation profiles between local and metastatic tumors in each subtypes of medulloblastoma?
Response 2:
We thank the reviewer for pointing this out.
As mentioned in the manuscript, metastatic disease was more commonly found at diagnosis in Group 3 medulloblastoma (MB) (25/38; 66%) followed by Group 4 MB (21/51; 41%). For SHH tumors, metastasis was detected in 6 out of 33 cases (18%). All WNT medulloblastoma (N= 15) had localized disease at diagnosis. We completed the data for WNT MB at diagnosis and made the changes in the text, page 3, line 103-104. The new sentence reads as follows:
“Metastatic disease was more commonly found at diagnosis in Group 3 (25/38; 66%) followed by Group 4 MB (21/51; 41%). For SHH tumors, metastasis was detected in 6 out of 33 cases (18%). All WNT MB had localized disease at diagnosis.”
Figure 2 describes the pattern of relapse according to molecular subgroups after receiving upfront radiotherapy-containing regimen and chemotherapy only

Reviewer 2 Report
Dear Editor,
thank you very much for giving me the opportunity to review this manuscript. The paper is well written and the scientific soundness is good.
I suggets only few comments and edits:
Page 2, line 85: “A total of 155 patients (83 boys and 72 84 girls) were enrolled into this retrospective study with a median follow-up of 6.6 years (range, 0.4-12.3 85 years).”
I wonder if a 0.4 years period of follow-up is too short to include patient in the statistics, in a population that is also numerically limited.
Page 3, line 107: replace ≠ with #
Page 6, figure 1B: The reviewer suggest to title the graph or to rename the ordinate on the graph
Page 7, line 154: replace ≠ with #
Page 8, line 157: remove extra spaces before “relapsed…”
Page 10, 184 & 185: “Only in one patient (patient ≠ 4), recurrence occurred on first-line treatment.” changes to “Recurrence occurred on first-line treatment only in one patient (patient # 4).”
Page 14, line 237: remove extra spaces before “survival…”
Page 14, lines 247-249 : “Establishing the impact of timing of relapse on OS post-relapse could allow us to guide appropriate surveillance for those who have completed treatment and to develop specific trials for children with relapsed medulloblastoma and test new therapies.”
I wonder if this statement is a bit of a stretch. It would be unobjectionable if more treatment options existed at relapse and clinicians were required to stratify risk and intensify treatment in early-relapsed versus late-relapsed patients. But in MBs, as accurately presented in the paper, treatment options are scarce and outcomes still unsatisfactory, so perhaps identifying recurrence early does not add any particular benefit at the present time.
Page 16, line 357: add a space after “[44].”
Page 19, line 409: “Casnova, M” changes to “Casanova, M”
Best regards
Author Response
Response to Reviewer 2 Comments
The authors would like to thank the reviewer for precise and thoughtful comments and constructive suggestions which has led to a better manuscript. Below we respond to each referee comments individually. Please note that lines and pages refer to the revised manuscript with track changes.
Point 1: Page 2, line 85: “A total of 155 patients (83 boys and 72 84 girls) were enrolled into this retrospective study with a median follow-up of 6.6 years (range, 0.4-12.3 85 years).” I wonder if a 0.4 years period of follow-up is too short to include patient in the statistics, in a population that is also numerically limited.
Response 1:
We agree with the reviewer that within a limited follow-up period (median follow-up was 6.6 years with a minimum follow-up of 0.44 years for one patient), we cannot provide any information on the eventual late-occurring events of these patients. Although, due to limited numbers of patients in this cohort, all patients were included irrespective of the length of their follow-up.
Secondly, the purpose of this study was to identify these patients with a relapsed or progressive disease and to look at the prognostic factors associated with survival post relapse or progression. As described in previous studies, early relapses occur within the first few months after the end of the treatment, especially in Group 3 medulloblastoma. This is a reason why we included these patients in our study cohort. Taken together, this study adds some information to previous data written about patients treated for a medulloblastoma and the risk of very early relapse.
Point 2: Page 3, line 107: replace ≠ with #
Response 2:
The suggested correction has been made, page 3, line 111.
Point 3: Page 6, figure 1B: The reviewer suggests to title the graph or to rename the ordinate on the graph
Response 3:
As suggested by the reviewer, we have revised Figure 1B by adding a title to both graphs: “Overall survival according to molecular subgroup” on Figure A and “Event free survival according to molecular subgroup” on Figure B.
Point 4: Page 7, line 154: replace ≠ with #
Response 4:
The suggested correction has been made to the text, page 7, line 177.
Point 5: Page 8, line 157: remove extra spaces before “relapsed…”
Response 5:
The suggested correction has been made to the text, page 8, line 182.
Point 6: Page 10, 184 & 185: “Only in one patient (patient ≠ 4), recurrence occurred on first-line treatment.” changes to “Recurrence occurred on first-line treatment only in one patient (patient # 4).
Response 6:
Thank you for your suggestion. We replaced ≠ with # before patient 4 (page 11, line 219) and specified it was one patient out of 11 children described. The sentence now reads (page 11, line 218-219):
“Recurrence occurred on first-line treatment only in one patient out of 11 (patient # 4).”
Point 7: Page 14, line 237: remove extra spaces before “survival…”
Response 7:
The suggested correction has been made to the text, page 14, line 280.
Point 8: Page 14, lines 247-249: “Establishing the impact of timing of relapse on OS post-relapse could allow us to guide appropriate surveillance for those who have completed treatment and to develop specific trials for children with relapsed medulloblastoma and test new therapies.”
I wonder if this statement is a bit of a stretch. It would be unobjectionable if more treatment options existed at relapse and clinicians were required to stratify risk and intensify treatment in early-relapsed versus late-relapsed patients. But in MBs, as accurately presented in the paper, treatment options are scarce and outcomes still unsatisfactory, so perhaps identifying recurrence early does not add any particular benefit at the present time.
Response 8:
We would like to thank the reviewer for pointing out this issue. Indeed, the auteurs agree with the reviewer that the timing of relapse will not add any particular benefit to the surveillance and different treatment options in current times. In consequence, we decided to remove this sentence, page 14, line 288.
Point 9: Page 16, line 357: add a space after “[44].”
Response 9:
The suggested correction has been made to the text, page 16, line 399.
Point 10: Page 19, line 409: “Casnova, M” changes to “Casanova, M”
Response 10:
The suggested correction has been made to the reference, page 19, line 542.
Reviewer 3 Report
Huybrechts et al. analyzed the clinico-pathological and outcome data of a large series of pediatric medulloblastomas focusing on variables potentially associated with outcome at disease relapse. The results are interesting and the paper is well-written. I just have some minor comments/suggestions which could improve the paper.
- Line 57: something is missing after clinicopathological (e.g. characteristics)
- Line 61: "ranges" should be used with "between" instead of "from"
- Line 67: "distinct" should go after "transcriptionally and genetically"
- Line 77: the comma should be removed after "Although"
- Lines 88-90: although, medulloblastoma with extensive nodularity is rare, the present series is quite large, so it is a bit strange that this subtype wasn't observed. I would specify in this paragraph that no medulloblastoma with extensive nodularity was observed, so that it is sure that this subtype was taken into consideration.
- Line 184-185: the sentence stating the only in one patient recurrence occurred on first-line treatment is unclear. Do you mean the first relapse? If it is so, I would move it previously when the general data regarding relapsed cases are described. If instead you mean one out of the 11 children described in the previous sentence, I would specify it.
- Lines 192-194: considered that 4 posterior fossa high-grade gliomas were identified and that only in 6 children disease recurrence was biopsied, isn't possible that in a subset of cases a second glioma was "misdiagnosed" as medulloblastoma recurrence? Could you add a comment to this topic? Would you suggest bioptic samples if feasible?
- Regarding second malignancies, do you have data regarding germline syndromes in your cohort?
- Lines 230-231 and 237-238: please check this sentence.
- Please consider to improve the quality of figures, especially of Figure 3.
Author Response
Response to Reviewer 3 Comments
The authors would like to thank the reviewer for precise and thoughtful comments and constructive suggestions which has led to a better manuscript. Below we respond to each referee comments individually. Please note that lines and pages refer to the revised manuscript with track changes.
Point 1: Line 57: something is missing after clinicopathological (e.g. characteristics)
Response 1:
Thank you for your suggestion. The sentence now reads (page 2, line 56-59):
“Treatment outcome is strongly associated with patient age and a series of established and evolving clinicopathological characteristics including metastatic disease, histology, postoperative residual disease (< or ³ 1.5 cm2), MYC amplification status and more recent molecular features [2-5].”
Point 2: Line 61: "ranges" should be used with "between" instead of "from"
Response 2:
Thank you for your suggestion. The sentence now reads (page 2, line 59-61):
“Five-year overall survival (OS) for standard-risk patients, typically defined as patients older than 3 years of age at diagnosis, who have a gross total resection of their tumor and are non-metastatic at diagnosis, ranges between 70-85% [6-9].”
Point 3: Line 67: "distinct" should go after "transcriptionally and genetically"
Response 3:
As suggested by the reviewer we moved the word “distinct” after "transcriptionally and genetically". The sentence now reads (page 2, line 64-66):
“More recent insights into the biology of MB have shown that this group of tumors comprises at least four distinct molecular subgroups, Wingless (WNT), Sonic Hedgehog (SHH), Group 3 and Group 4, with transcriptionally and genetically distinct profiles and correlated clinical outcome.”
Point 4: Line 77: the comma should be removed after "Although"
Response 4:
The correction has been made on page 2, line 77.
Point 5: Lines 88-90: although, medulloblastoma with extensive nodularity is rare, the present series is quite large, so it is a bit strange that this subtype wasn't observed. I would specify in this paragraph that no medulloblastoma with extensive nodularity was observed, so that it is sure that this subtype was taken into consideration.
Response 5:
As suggested by the reviewer the following sentence was added to the text, page 2, line 90-91:
“No medulloblastoma with extensive nodularity were identified in this cohort.”
Point 6: Line 184-185: the sentence stating the only in one patient recurrence occurred on first-line treatment is unclear. Do you mean the first relapse? If it is so, I would move it previously when the general data regarding relapsed cases are described. If instead you mean one out of the 11 children described in the previous sentence, I would specify it.
Response 6:
Thank you for your comment. As suggested the sentence page 11, line 218-219 has been rephrased as follows:
“Recurrence occurred on first-line treatment only in one patient out of 11 (patient # 4).”
Point 7: Lines 192-194: considered that 4 posterior fossa high-grade gliomas were identified and that only in 6 children disease recurrence was biopsied, isn't possible that in a subset of cases a second glioma was "misdiagnosed" as medulloblastoma recurrence? Could you add a comment to this topic? Would you suggest bioptic samples if feasible?
Response 7:
We highly appreciate the reviewer comment regarding the need of a re-biopsy at relapse.
Indeed, due to the spatial and temporal heterogeneity of medulloblastoma, there is a need to integrate patients with a recurrent medulloblastoma in more personalized medicine approaches, which recommend a biopsy of the tumor at recurrence. Moreover, recent studies have shown that despite subgroup affiliation being preserved at recurrence and between both primary and metastatic components, there is a substantial divergence of the dominant clone at recurrence (studies of Hill et al, Morrisy et al, Ramaswamy et al).
Therefore, we made some changes to the text by adding and correcting following sentences:
- Page 6, line 138-140: “A biopsy or surgical resection of the tumor was performed in case of focal relapse, in the context or molecular screening programs (precision medicine in the field of pediatric oncology) or in ambiguous cases.”
- Page 7, line 174-175: “Fifteen out of 48 patients were able to undergo resection or biopsy of the relapsing tumor which involved local recurrence in 9 cases and metastatic sites in 6.”
- Page 11, line 219: “A relapse was confirmed by biopsy in 6 out of 11 cases.”
Point 8: Regarding second malignancies, do you have data regarding germline syndromes in your cohort?
Response 8:
I would like to thank the reviewer for his comment.
All 6 patients have been referred for genetic counseling and germline genetic testing has been performed. Among the 6 patients, only one patient had a germline TP53 mutation. As such, we added the following sentence to the text, page 11, line 228-229:
“These patients have been referred for genetic counseling and underwent germline testing. Of the 6 patients, only one was found to carry a germline TP53 mutation.”
Point 9: Lines 230-231 and 237-238: please check this sentence.
Response 9:
We thank the reviewer for pointing this out and revised the text as follows:
- The sentence on line 230 – 231, now reads on page 11, line 259-262: “The percentage of selection of salvage RT at first relapse varied between 39% to 59%, and that of the time between diagnosis and first relapse was less than 56%. As such, no definitive conclusion could be drawn about the prognostic value of both explanatory variables on OS post-relapse (Table 3). “
- We removed the space between post-relapse and survival and rewrote the sentence as follows (page 14, line 277-280): “This large retrospective study, focusing on putative clinico-molecular factors associated with post-relapse survival in childhood, identified molecular subgrouping at diagnosis to be significantly related to survival post-relapse.”
Point 10: Please consider to improve the quality of figures, especially of Figure 3.
Response 10:
- As suggested by the reviewer, all figures (1-5), and especially figure 3 as well as table 1 have been revised and modified with a higher resolution.
- To the editor: In line with the suggestions of the second reviewer (figure 1B) we added following title “Overall survival after progression/relapse “into figure 5.